# Stereoscopic Virtual Reality Teleoperation for Human Robot Collaborative Dataset Collection

Yi-Shiuan Tung*
Matthew B. Luebbers*
yi-shiuan.tung@colorado.edu
matthew.luebbers@colorado.edu
University of Colorado Boulder
Boulder, Colorado, USA

Alessandro Roncone
alessandro.roncone@colorado.edu
University of Colorado Boulder
Lab0 Inc.
Boulder, Colorado, USA

Bradley Hayes
bradley.hayes@colorado.edu
University of Colorado Boulder
Boulder, Colorado, USA

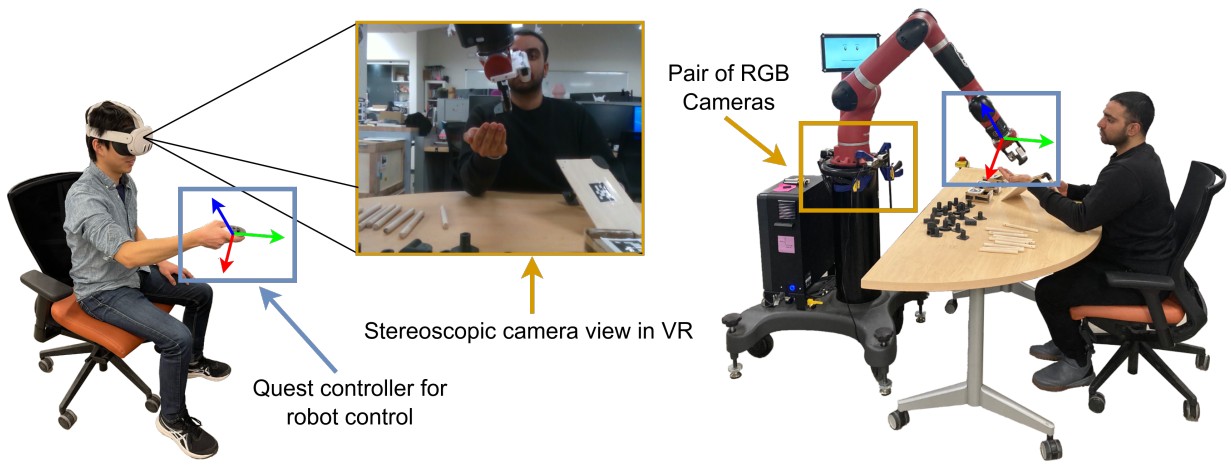

Pair of RGB Cameras

Stereoscopic camera view in VR

Quest controller for robot control

Figure 1: Our virtual reality (VR) teleoperation system projects a stereoscopic camera view to the VR headset, providing the human operator an egocentric perspective and a detailed rendering of the environment with depth perception. The human operator manipulates the robot end effector by moving and controlling the Quest 3 controller. We intend to use our system for collecting data where robots collaborate with humans on various tasks.

## ABSTRACT

Large and diverse datasets are required to train general purpose models in NLP, computer vision, and robot manipulation. However, existing robotics datasets have single robots interacting in a static environment whereas in many real world scenarios, robots have to interact with humans or other dynamic agents. In this work, we present a virtual reality (VR) teleoperation system to enable data collection for human robot collaborative (HRC) tasks. The human operator using the VR system receives an immersive and high fidelity egocentric view with a stereoscopic depth effect, providing the situational awareness required to teleoperate the robot remotely to perform various tasks. We propose to collect data on a set of HRC tasks and introduce a taxonomy to categorize the tasks. We envision that our VR system will broaden the scope of tasks robots can

perform with human collaborators and that the proposed dataset will enable the development of new algorithms for HRC.

## CCS CONCEPTS

• **Human-centered computing → Virtual reality**; **Collaborative interaction**.

## KEYWORDS

virtual reality, teleoperation, human robot collaboration

**ACM Reference Format:**
Yi-Shiuan Tung, Matthew B. Luebbers, Alessandro Roncone, and Bradley Hayes. 2024. Stereoscopic Virtual Reality Teleoperation for Human Robot Collaborative Dataset Collection. In *HRI '24: Virtual, Augmented and Mixed-Reality for Human-Robot Interactions Workshop, March 11, 2024, Boulder, CO*. ACM, New York, NY, USA, 4 pages.

## 1 INTRODUCTION AND MOTIVATION

Large-scale pretrained models that are trained on broad and diverse datasets have shown generalizability and adaptability across multiple tasks in various environments. The Open X-Embodiment [14], which compiles robotic manipulation datasets from different sources and robot embodiments, has demonstrated the effectiveness of transformer-based models trained on this data. While these

---

*Both authors contributed equally to this research.

datasets enable robots to learn generalizable skills in a variety of settings, the robots are interacting in mostly static environments without humans or other dynamic agents. As robots get deployed in homes and public spaces, these autonomous agents must learn to interact or collaborate with humans. Importantly, humans and robots have complementary skills: humans have strong reasoning abilities and adaptability, whereas robots excel in numerical tasks and precision. This creates a synergistic partnership, making human robot collaboration an advantageous approach for many tasks [20].

Teleoperating robots protects the human operator from hazardous environments and also enables data collection on tasks demanding human dexterity, expertise, and extensive background knowledge, all without the need for the human to be physically present [6]. Traditional systems display camera video streams on a computer monitor and rely on keyboards or joysticks to control the robot. On the other hand, virtual reality (VR) interfaces offer an immersive 3D experience, enabling the user to perceive depth and translate human arm movement to robot actions.

In this paper, we present a VR teleoperation system intended for collecting data on human-robot collaborative tasks. We introduce a series of shared workspace tasks along with a taxonomy that indicates if the task involves shared contact, whether the action space of the human and the robot are the same (homogeneous) or different (heterogeneous), and if the robot assumes a leader or a supporter role. We envision that such a dataset will facilitate the development of robot learning algorithms that collaborate with humans on various tasks.

## 2 RELATED WORK

Virtual reality (VR) interfaces provide an immersive 3D environment for better situational awareness and a more intuitive method for robot control compared to traditional teleoperation systems that use monitors and keyboards [19]. VR teleoperation systems that remotely control a robot have been developed in domains ranging from space [15] to surgery [17] to manufacturing [5, 10]. To visualize the environment, VR headsets often render a point cloud from remote color and depth cameras [12, 16]. Omarali et al. [13] uses a RGBD camera to render an OctoMap mapping of the remote environment which has fewer distortions and occlusions compared to point clouds. Wei et al. [18] uses a stereo camera and aligns a local camera on the robot end effector to the global 3D point cloud. Our system uses a pair of RGB cameras, with one casting its feed to the VR interface's left eye and the other to the right, creating a perception of depth (stereopsis) for the human operator [9]. The cameras are positioned on the robot's body to provide an egocentric view, allowing the human operator to provide controls from the robot's perspective. Using stereo camera data delivers the highest fidelity reconstruction of the environment possible, with greater accuracy than can be achieved with point cloud methods, at the expense of limiting the operator's viewpoint to that of the camera.

VR teleoperation is a popular choice for collecting data from robots, and prior work has collected large scale datasets of robots manipulating objects [7, 22]. When trained using imitation learning, robots have demonstrated high success rates and generalization. However, the vast majority of existing datasets only include single robot tasks in static environments whereas many real world tasks involve interaction with humans or other dynamic agents. In this work, we present a VR teleoperation system for collecting data on robots collaborating with humans.

Previous research has collected datasets on human robot interaction that have facilitated robot learning and the learning of human behavior models. Ben-Youssef et al. [2] recorded humans interacting with the social robot Pepper. Celiktutan et al. [3] introduced human-human and human-human-robot datasets where participants asked personality questions to each other. Some works have provided multimodal datasets where a human teaches a robot to recognize new objects [1, 8]. In contrast, our proposed data collection focuses on physical human robot collaborative tasks, as summarized in Table 1.

## 3 VR INTERFACE DESIGN

### 3.1 Stereoscopic Visualization

To achieve an immersive, 3D visualization of the environment from the robot's point of view, we pass dual RGB camera feeds to a VR interface. In our case, we use a pair of RealSense D435 cameras, communicating with a Meta Quest 3 headset. The cameras are placed next to each other, spaced roughly to match an average human's interpupillary distance. The feed from the leftmost camera is passed to the left eye of the operator, with the rightmost camera passed to the right eye. The binocular disparity in these images creates a depth effect in the viewer, tricking the visual cortex to interpret the scene as 3-dimensional [4].

Since the camera position and orientation are not tied to the head movements of the operator, the camera feeds are projected onto a spatially-anchored window within the immersive VR environment, almost as if the operator were looking at a large monitor displaying the robot's camera feed in 3D. This prevents any motion sickness in operators that would arise from moving their head and not seeing a corresponding motion in their environment, causing a mismatch between the senses of vision and proprioception [11].

### 3.2 Teleoperation

Operators are able to command the robot using Meta Quest Touch Plus handheld controllers within the VR interface (a single controller for stationary manipulators and two controllers for mobile manipulator robots). For controlling the base of a mobile robot, operators use a pair of thumb sticks to control robot translation and rotation. For controlling a manipulator arm, we use the 6DOF position of the right hand controller, collected via the VR headset's internal tracking, with the human spatially tracing intended behavior for the robot's end effector.

To prevent unwanted robot arm movement when the operator is not engaged in a manipulation task, controller poses are only passed through to the robot's inverse kinematic solver when the trigger button on the right hand controller is depressed. When the operator does not have the trigger depressed, a semi-transparent hologram of the controller is displayed, positioned in the immersive VR coordinate system so that it maps to the current position and orientation of the real robot's end effector. When the operator wishes to begin control of the manipulator arm, they will match their own controller position with that of the semi-transparent hologram,

| Shared Contact, Homogeneous Actions | Shared Contact, Heterogeneous Actions | Non-Shared Contact, Homogeneous Actions | Non-Shared Contact, Heterogeneous Actions |
|---|---|---|---|
| 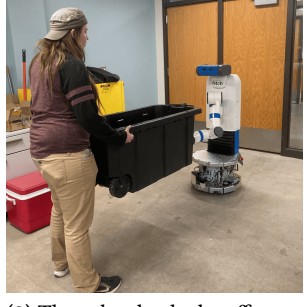 | 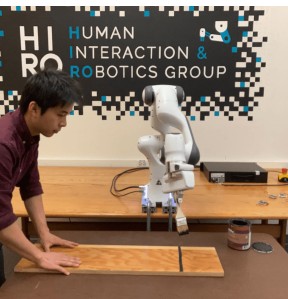 | 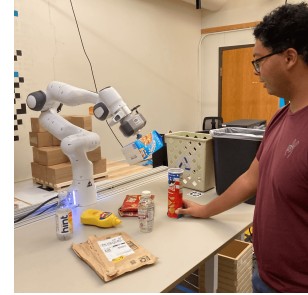 | 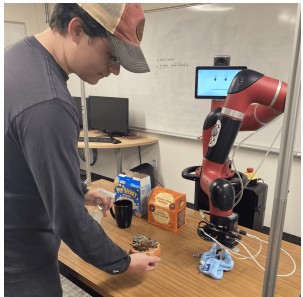 |
| **(2)** The robot leads the effort to move the box through the door. | **(3)** The robot paints the wood while the human stabilizes it. | **(4)** The robot sorts recyclables while the human assists. | **(5)** The human lifts up objects as the robot cleans the table. |
| 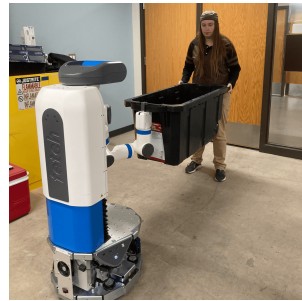 | 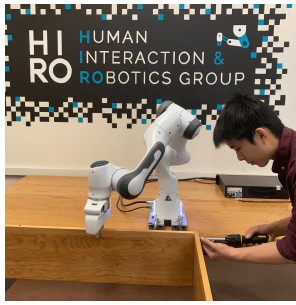 | 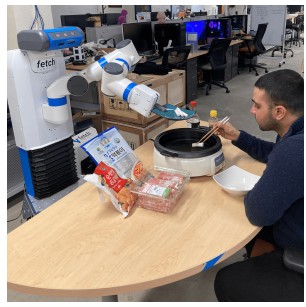 | 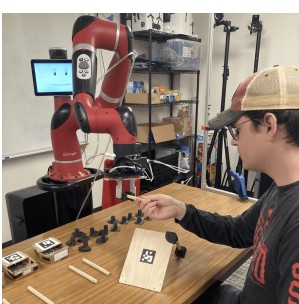 |
| **(6)** The robot follows the human's lead moving the box. | **(7)** The robot holds the board while the human inserts screw. | **(8)** The robot cooks the food for the human to consume. | **(9)** The robot brings parts to assist in the assembly. |

*(Leader — top row; Supporter — bottom row)*

**Table 1: The taxonomy of our proposed human robot collaboration tasks for data collection.**

depress the trigger, and begin their movement. This prevents large uncontrolled robot motions interpolating between discontinuous poses on either end of a gap in teleoperation. Operators can open or close the gripper with a button reachable by the right thumb.

## 4 PROPOSED DATA COLLECTION

Using the VR teleoperation interface, the human operator sitting in a remote location can control a robot that is collocated with human participants to collaborate on a variety of tasks. We plan to collect a human-robot interaction dataset, recording robot joint states, the robot's camera view (also the human operator's view through VR), and third person camera views that capture the robot and the human in the same frame. The human participants will wear arm sleeves, gloves, and a vest with fiducial markers tracked via a set of OptiTrack motion capture cameras, providing ground truth human poses. We will also include natural language text descriptions of the human and robot's tasks. We categorize our proposed human-robot collaboration tasks via the following taxonomy: 1) **Shared Contact vs. Non-Shared Contact**: the human and the robot interact with the same vs. different objects, 2) **Homogeneous vs. Heterogeneous Actions**: the human and the robot have the same vs. different action spaces, and 3) **Leader vs. Supporter Roles**: the human adapts to the robot's actions vs. the robot adapts to

the human. Some tasks allow the robot to take either the leader or supporter role, and we plan to collect data capturing the robot's behavior for both cases.

### 4.1 Shared Contact

*4.1.1 Homogeneous Actions.* The task requires the human robot team to carry heavy or large and unwieldy objects such as boxes, furniture or planks of wood for construction. For **leader role**, the robot guides the human towards the destination (Fig. 2) while the robot follows the human in the **supporter role** (Fig. 6).

*4.1.2 Heterogeneous Actions.* The human and the robot are interacting with the same object but perform different actions. For example, the robot in a **leader role** paints the wood while the human stabilizes it (Fig. 3). In a **supporter role**, the robot stabilizes a wooden plank while the human inserts screw (Fig. 7).

### 4.2 Non-Shared Contact

*4.2.1 Homogeneous Actions.* The human and the robot are sorting recyclables into the correct bins (Fig. 4). The robot assumes the **leader role** by sorting recyclables while the human supervises and assists with items placed in the wrong bins or items the robot cannot pick up. In a **supporter role**, the robot maintains a belief of the item the human is picking up and selects a different item to

sort. Another **supporter role** task is a robotic chef that places food into a hot pot to cook and also picks up cooked food for the human (Fig. 8).

*4.2.2 Heterogeneous Actions.* The human robot team is tasked to clean the table (Fig. 5). In a **leader role**, the robot wipes the surface with a cloth while the human lifts up objects on the table to allow the robot to clean the area beneath the objects. Conversely, when the robot adopts a **supporter role**, it takes on the task of lifting objects and the human wipes down the surfaces. The second task is a collaborative assembly of a miniature table [21]. The robot takes on a **supporter role** and fetches parts for the human as the human assembles the table (Fig. 9).

## 5 CONCLUSION

In this paper, we introduce a VR teleoperation system to collect data on a robot collaborating with humans. Instead of displaying point clouds in VR, our approach involves streaming data from two RGB cameras onto a plane in Unity to create a high fidelity reconstruction of the environment along with depth perception from stereopsis. We implement an intuitive interface for controlling the robot, directly translating human arm movements to robot end effector motion and using the Quest controller thumbsticks for translation and rotation of mobile bases. Lastly, we present a taxonomy of human robot collaboration tasks and provide examples for each categorization from which we aim to gather data.

We envision that our VR teleoperation system will enable dexterous manipulation of objects in complex environments, broadening the scope for robots to engage in more sophisticated tasks alongside humans. We plan to make the dataset publicly available, facilitating the development of robot learning that collaborates with humans and other agents.

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

## ACKNOWLEDGMENTS

This work was supported by the Office of Naval Research under Grant N00014-22-1-2482 and the Army Research Laboratory under Grant W911NF-21-2-0126. The authors would like to thank Wei Jiang for his assistance with image editing.