# OpenReview forum: "Stereoscopic Virtual Reality Teleoperation for Human Robot Collaborative Dataset Collection"
_humanrobotinteraction.org/HRI/2024/Workshop/VAM-HRI — VAM-HRI 2024 Oral_

### Official Review · Reviewer_qxXP · 2024-02-23
**Accept**

**Rating:** 8
**Confidence:** 4

**Review:**

This paper presents a VR teleoperation system that is to be used for collecting a dataset on human-robot collaboration tasks. These tasks include shared contact vs. non-shared contact, homogenous vs heterogeneous actions, and leader vs. supporter roles. The authors plan to release this dataset publicly in the future.

Strengths:

- This paper utilizes known VR visualization techniques to provide a clearer teleoperation interface for robots. It provides a clearer picture than 3D point clouds making it a particularly useful visualization modality for teloperation.

- The paper plans to collect a dataset from multiple robots performing a variety of tasks. If successful, this dataset will be an important contribution to the robotics community. I look forward to seeing future work based on this dataset.

Area for improvement:

- Will the joint data points be filtered? For example, as a robot follows a user’s hand, there will be random movement to the user’s hand. This may result in rough waypoints that will need to be smoothed out somehow.

---

### Official Review · Reviewer_Xpd9 · 2024-02-26
**reviewer B**

**Rating:** 8
**Confidence:** 5

**Review:**

The paper presents a VR framework for data collection for teleoperation in HRI. The authors considered multiple tasks and scenarios. The paper is solid but could use some clarification.

 It would be interesting to see what exactly is being collected: hands tracking, images, eye tracking, etc. It feels a bit incomplete without that information.

Regarding operator pose and hand tracking, if you are collecting it, are you using the internal localization of the headset/controllers or mocap? It would be interesting to compare them.

Following on that, to my understanding there are always 2 people involved (robot operator and participant) + robot. It would be again interesting to track as many things as possible on both participants and operators, not sure if the authors considered that.

Overall, accept!

---

### Decision · Program_Chairs · 2024-02-26

Accept (Oral)